# Detrended Partial Cross Correlation for Brain Connectivity Analysis

**Jaime S Ide**[*]
Yale University
New Haven, CT 06519
jaime.ide@yale.edu

**Fabio A Cappabianco**
Federal University of Sao Paulo
S.J. dos Campos, 12231, Brazil
cappabianco@unifesp.br

**Fabio A Faria**
Federal University of Sao Paulo
S.J. dos Campos, 12231, Brazil
ffaria@unifesp.br

**Chiang-shan R Li**
Yale University
New Haven, CT
chiang-shan.li-yale.edu

## Abstract

Brain connectivity analysis is a critical component of ongoing human connectome projects to decipher the healthy and diseased brain. Recent work has highlighted the power-law (multi-time scale) properties of brain signals; however, there remains a lack of methods to specifically quantify short- vs. long- time range brain connections. In this paper, using detrended partial cross-correlation analysis (DPCCA), we propose a novel functional connectivity measure to delineate brain interactions at multiple time scales, while controlling for covariates. We use a rich simulated fMRI dataset to validate the proposed method, and apply it to a real fMRI dataset in a cocaine dependence prediction task. We show that, compared to extant methods, the DPCCA-based approach not only distinguishes short and long memory functional connectivity but also improves feature extraction and enhances classification accuracy. Together, this paper contributes broadly to new computational methodologies in understanding neural information processing.

## 1 Introduction

Brain connectivity is crucial to understanding the healthy and diseased brain states [15, 1]. In recent years, investigators have pursued the construction of human connectomes and made large datasets available in the public domain [23, 24]. Functional Magnetic Resonance Imaging (fMRI) has been widely used to examine complex processes of perception and cognition. In particular, functional connectivity derived from fMRI signals has proven to be effective in delineating biomarkers for many neuropsychiatric conditions [15].

One of the challenges encountered in functional connectivity analysis is the precise definition of nodes and edges of connected brain regions [21]. Functional nodes can be defined based on activation maps or with the use of functional or anatomical atlases. Once nodes are defined, the next step is to estimate the weights associated with the edges. Traditionally, these functional connectivity weights are measured using correlation-based metrics. Previous simulation studies have shown that they can be quite successful, outperforming higher-order statistics (e.g. linear non-gaussian acyclic causal models) and lag-based approaches (e.g. Granger causality) [20].

On the other hand, very few studies have investigated the power-law cross-correlation properties (equivalent to multi-time scale measures) of brain connectivity. Recent research suggested that fMRI

---

[*]Corresponding author: Department of Psychiatry, 34 Park St. S110. New Haven CT 06519.

signals have power-law properties (e.g. their power-spectrum follows a power law) [8, 3] and that the deviations from the typical range of power-exponents have been noted in neuropsychiatric disorders [11]. For instance, in [3], using wavelet-based multivariate methods, authors observed that scale-free properties are characteristic not only of univariate fMRI signals but also of pairwise cross-temporal dynamics. Moreover, they found an association between the magnitude of scale-free dynamics and task performance. We hypothesize that power-law correlation measures may capture additional dimensions of brain connectivity not available from conventional analyses and thus enhance clinical prediction.

In this paper, we aim to answer three key open questions: (i) whether and how brain networks are cross-correlated at different time scales with long-range dependencies ("long-memory" process, equivalent to power-law in the frequency domain); (ii) how to extract the intrinsic association between two regions controlling for the influence of other interconnected regions; and (iii) whether multi-time scale connectivity measures can improve clinical prediction. We address the first two questions by using the detrended partial cross-correlation analsyis (DPCCA) coefficient [25], a measure that quantifies correlations on multiple time scales between non-stationary time series, as is typically the case with task-related fMRI signals. DPCCA is an extension of detrended cross-correlation analysis [17, 13], and has been successfully applied to analyses of complex systems, including climatological [26] and financial [18] data. Unlike methods based on filtering particular frequency bands, DPCCA directly informs correlations across multiple time scales, and unlike wavelet-based approaches (e.g. cross wavelet transformation and wavelet transform coherence [2]), DPCCA has the advantage of estimating pairwise correlations controlling for the influence of other regions. This is critical because brain regions and thus fMRI signals thereof are highly interconnected. To answer the third question, we use the correlation profiles, generated from DPCCA, as input features for different machine learning methods in classification tasks and compare the performance of DPCCA-based features with all other competing features.

In Section 2, we describe the simulated and real data sets used in this study, and show how features of the classification task are extracted from the fMRI signals. In Section 3, we provide further details about DPCCA (Section 3.1), and present the proposed multi-time scale functional connectivity measure (Section 3.2). In Section 4, we describe core experiments designed to validate the effectiveness of DPCCA in brain connectivity analysis and clinical prediction. We demonstrate that DPCCA (i) detects connectivity at multiple-time scales while controlling for covariates (Sections 4.1 and 4.3), (ii) accurately identifies functional connectivity in well-known gold-standard simulated data (Section 4.2), and (iii) improves classification accuracy of cocaine dependence with fMRI data of seventy-five cocaine dependent and eighty-eight healthy control individuals (Section 4.4). In Section 5, we conclude by highlighting the significance of the study as well as the limitations and future work.

## 2 Material and Methods

### 2.1 Simulated dataset: NetSim fMRI data

We use fMRI simulation data - NetSim [20] - previously developed for the evaluation of network modeling methods. Simulating rich and realistic fMRI time series, NetSim is comprised of twenty-eight different brain networks, with different levels of complexity. These signals are generated using dynamic causal modeling (DCM [6]), a generative network model aimed to quantify neuronal interactions and neurovascular dynamics, as measured by the fMRI signals. NetSim graphs have 5 to 50 nodes organized with "small-world" topology, in order to reflect real brain networks. NetSim signals have 200 time points (mostly) sampled with repetition time (TR) of 3 seconds. For each network, 50 separate realizations ("subjects") are generated. Thus, we have a total of 1400 synthetic dataset for testing. Finally, once the signals are generated, white noise of standard deviation 0.1-1% is added to reproduce the scan thermal noise.

### 2.2 Real-world dataset: Cocaine dependence prediction

Seventy-five cocaine dependent (CD) and eighty-eight healthy control (HC) individuals matched in age and gender participated in this study. CD were recruited from the local, greater New Haven area in a prospective study and met criteria for current cocaine dependence, as diagnosed by the Structured Clinical Interview for DSM-IV. They were drug-free while staying in an inpatient treatment unit.

The Human Investigation committee at Yale University School of Medicine approved the study, and all subjects signed an informed consent prior to participation. In the MR scanner, they performed a simple cognitive control paradigm called stop-signal task [14]. FMRI data were collected with 3T Siemens Trio scanner. Each scan comprised four 10-min runs of the stop signal task. Functional blood oxygenation level dependent (BOLD) signals were acquired with a single-shot gradient echo echo-planar imaging (EPI) sequence, with 32 axial slices parallel to the AC-PC line covering the whole brain: TR=2000 ms, TE=25 ms, bandwidth=2004 Hz/pixel, flip angle=85°, FOV=220×220 $mm^2$, matrix=66×64, slice thickness=4 mm and no gap. A high-resolution 3D structural image (MPRAGE; 1 mm resolution) was also obtained for anatomical co-registration. Three hundred images were acquired in each session. Functional MRI data was pre-processed with standard pipeline using Statistical Parametric Mapping 12 (SPM12) (Wellcome Department of Imaging Neuroscience, University College London, U.K.).

### 2.2.1 Brain activation

We constructed general linear models and localized brain regions responding to conflict (stop signal) anticipation (encoded by the probability P(stop)) at the group level [10]. The regions responding to P(stop) comprised the bilateral parietal cortex, the inferior frontal gyrus (IFG) and the right middle frontal gyrus (MFG); and regions responding to motor slowing bilateral insula, the left precentral cortex (L.PC), and the supplementary motor area (SMA) (Fig. 1(a))[2]. These regions of interest (ROIs) were used as masks to extract average activation time courses for functional connectivity analyses.

### 2.2.2 Functional connectivity

We analyzed the frontoparietal circuit involved in conflict anticipation and response adjustment using a standard Pearson correlation analysis and multivariate Granger causality analysis or mGCA [19]. In Fig. 1(b), we illustrate fifteen correlation coefficients derived from the six ROIs for each individual CD and HC as shown in Fig. 1(a). According to mGCA, connectivities from bilateral parietal to L.PC and SMA were disrupted in CD (Fig. 1(b)). These findings offer circuit-level evidence of altered cognitive control in cocaine addiction.

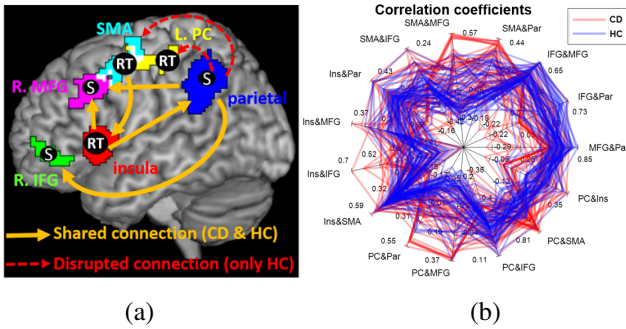

(a)                                                         (b)

Figure 1: Disrupted frontoparietal circuit in cocaine addicts. The frontoparietal circuit included six regions responding to Bayesian conflict anticipation ("S") and regions of motor slowing ("RT"): (a) CD and HC shared connections (orange arrows). (b) Connectivity strengths between nodes in the frontoparietal circuit. We show connectivity strengths between nodes for each individual subject in CD (red line) and HC (blue line) groups.

## 3 A Novel Measure of Brain Functional Connectivity

### 3.1 Detrended partial cross-correlation analysis (DPCCA)

Detrended partial cross-correlation is a novel measure recently proposed by [25]. DPCCA combines the advantages of detrended cross-correlation analysis (DCCA) [17] and standard partial correlation. Given two time series $\{x^{(a)}\}, \{x^{(b)}\} \in X_t$, where $X_t \in \mathbb{R}^m, t = 1, 2, ..., N$ time points, DPCCA is given by Equation 1:

$$\rho_{DPCCA}(a, b; s) = \frac{-C_{a,b}(s)}{\sqrt{C_{a,a}(s).C_{b,b}(s)}}, \tag{1}$$

where $s$ is the time scale and each term $C_{a,b}(s)$ is obtained by inverting the matrix $\rho(s)$, e.g. $C(s) = \rho^{-1}(s)$. The coefficient $\rho_{a,b} \in \rho(s)$ is the so called DCCA coefficient [13]. The DCCA coefficient is an extension of the detrented cross correlation analysis [17] combined with detrended fluctuation analysis (DFA) [12].

Given two time series $\{x\}, \{y\} \in X_t$ (indices omitted for the sake of simplicity) with $N$ time points and time scale $s$, DCCA coefficient is given by Equation 2:

$$\rho(s) = \frac{F^2_{DCCA}(s)}{F_{DFA,x}(s)F_{DFA,y}(s)}, \tag{2}$$

where the numerator and denominator are the average of detrended covariances and variances of the $N - s + 1$ windows (partial sums), respectively, as described in Equations 3-4:

$$F^2_{DCCA}(s) = \frac{\sum_{j=1}^{N-s+1} f^2_{DCCA}(s,j)}{N-s} \tag{3}$$

$$F^2_{DFA,x}(s) = \frac{\sum_{j=1}^{N-s+1} f^2_{DFA,x}(s,j)}{N-s}. \tag{4}$$

The partial sums (profiles) are obtained with sliding windows across the integrated time series $X_t = \sum_{i=1}^{t} x_i$ and $Y_t = \sum_{i=1}^{t} y_i$. For each time window $j$ with size $s$, detrended covariances and variances are computed according to Equations 5-6:

$$f^2_{DCCA}(s,j) = \frac{\sum_{t=j}^{j+s-1} (X_t - \widehat{X_{t,j}})(Y_t - \widehat{Y_{t,j}})}{s-1}, \tag{5}$$

$$f^2_{DFA,x}(s,j) = \frac{\sum_{t=j}^{j+s-1} (X_t - \widehat{X_{t,j}})^2}{s-1}, \tag{6}$$

where $\widehat{X_{t,j}}$ and $\widehat{Y_{t,j}}$ are polynomial fits of time trends. We used a linear fit as originally proposed [13], but higher order fits could also be used [25]. DCCA can be used to measure power-law cross-correlations. However, we focus on DCCA coefficient as a robust measure to detect pairwise cross-correlation in multiple time scales, while controlling for covariates. Importantly, DPCCA quantifies correlations among time series with varying levels of non-stationarity [13].

### 3.2 DPCCA for functional connectivity analysis

In this section, we propose the use of DPCCA as a novel measure of brain functional connectivity. First, we show in simulation experiments that the measure satisfies desired connectivity properties. Further, we define the proposed connectivity measure. Although these properties are expected by mathematical definition of DPCCA, it is critical to confirm its validity on real fMRI data. Additionally, it is necessary to establish the statistical significance of the computed measures at the group level.

#### 3.2.1 Desired properties

Given real fMRI signals, the measure should accurately detect the time scale in which the pairwise connections occur, while controlling for the covariates. To verify this, we create synthetic data by combining real fMRI signals and sinusoidal waves (Fig. 2). To simplify, we assume additive property of signals and sinusoidal waves reflecting the time onset of the connections. For each simulation, we randomly sample 100 sets of time series or "subjects".

a) *Distinction of short and long memory connections*. Given two fMRI signals $\{x_A\}, \{x_B\}$, we derive three pairs with known connectivity profiles: short-memory $\{X_A = x_A + sin(T_1) + e\}$, $\{X_B = x_B + sin(T_1) + e\}$, long-memory $\{X_A = x_A + sin(T_2) + e\}$, $\{X_B = x_B + sin(T_2) + e\}$ and mixed $\{X_A = x_A + sin(T_1) + sin(T_2) + e\}$, $\{X_B = x_B + sin(T_1) + sin(T_2) + e\}$, where $T_1 << T_2$ and $e$ is a Gaussian signal to simulate measurement noise. We hypothesize that the two nodes A and B are functionally connected at time scales $T_1$ and $T_2$.

b) *Control for covariates.* Given three fMRI signals $\{x_A\}$, $\{x_B\}$, $\{x_C\}$, we derive three signals with known connectivity $\{X_{AC} = x_A + x_C + sin(T) + e\}$, $\{X_{BC} = x_B + x_C + sin(T) + e\}$, $\{X_C = x_C + e\}$, where $e$ is the measurement noise. We hypothesize that the two nodes A and B are functionally connected mostly at scale T, once the mutual influence of node C is controlled.

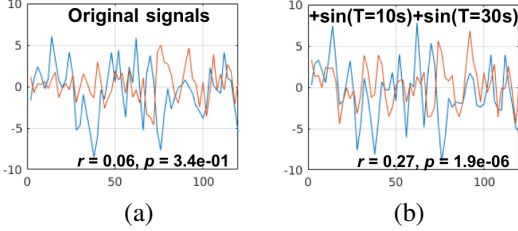

(a)                  (b)

Figure 2: Illustration of synthetic fMRI signals generated by combining real fMRI signals and sinusoidal waves. (a) Original fMRI signals, (b) original signals with $sin(T = 10s)$ and $sin(T = 30s)$ waves added.

### 3.2.2 Statistical significance

Given two nodes and their time series, we assume that they are functionally connected if the $\max |\rho_{DPCCA}|$, within a time range $s_{range}$, is significantly greater than the null distribution. Empirical null distributions are estimated from the original data by randomly shuffling time series across different subjects and nodes, as proposed in [20]. In this way, we generate realistic distributions of connectivity weights occurring by chance. Since we have a multivariate measure, the null dataset is always generated with the same number of nodes as the tested network. Multiple comparisons are controlled by estimating the false discovery rate. Importantly, the null distribution is also computed on $\max |\rho_{DPCCA}|$ within the time range $s_{range}$. We use a $s_{range}$ from 6 to 18 seconds, assuming that functional connections transpire in this range. Thus, we allow connections with different time-scales. We use this binary definition of functional connectivity for the current approach to be comparable with other methods, but it is also possible to work with the whole temporal profile of $\rho_{DPCCA}(s)$, as is done in the classification experiment (Section 4.4). To keep the same statistical criteria, we also generate null distributions for all the other connectivity measures.

### 3.2.3 DPCCA + Canonical correlation analysis

As further demonstrated by simulation results (Table 1), DPCCA alone has lower true positive rate (TPR) compared to other competing methods, likely because of its restrictive statistical thresholds. In order to increase the sensitivity of DPCCA, we augmented the method by including an additional canonical correlation analysis (CCA) [7]. CCA was previously used in fMRI in different contexts to detect brain activations [5], functional connectivity [27], and for multimodal information fusion [4]. In short, given two sets of multivariate time series $\{X_A(t) \in \mathbb{R}^m, t = 1, 2, ..., N\}$ and $\{X_B(t) \in \mathbb{R}^n, t = 1, 2, ..., N\}$, where $m$ and $n$ are the respective dimensions of the two sets $A$ and $B$, and $N$ is the number of time points, CCA seeks the linear transformations $u$ and $v$ so that the correlation between the linear combinations $X_A(t)u$ and $X_B(t)v$ is maximized. In this work, we propose the use of CCA to define the existence of a true connection, in addition to the DPCCA connectivity results. The proposed method is summarized in **Algorithm 1**. With CCA (Lines 8-14), we identify the nodes that are strongly connected after linear transformations. In Line 18, we use CCA to inform DPCCA in terms of positive connections.

## 4 Experiments and Results

### 4.1 Connectivity properties: Controlling time scales and covariates

In Figure 3, we observe that DPCCA successfully captured the time scales of the correlations between time series $\{X_A\}$, $\{X_B\}$, despite the noisy nature of fMRI signals. For instance, it distinguished between short and long-memory connections, represented using $T_1 = 10s$ and $T_2 = 30s$, respectively (Figs. 3a-c). Importantly, it clearly detected the peak connection at $10s$ after controlling for the influence of covariate signal $X_C$ (Fig. 3f). Further, unlike DPCCA, the original DCCA method did not rule out the mutual influence of $X_C$ with peak at $30s$ (Fig. 3e).

**Algorithm 1** DPCCA+CCA

**Input:** Time series $\{X_t \in \mathbb{R}^m, t = 1, 2, ..., N\}$, where $m$ is the number of vectors and $N$ is the number of time points; time range $s_{range}$ with $k$ values
**Output:** Connectivity matrix $FC : [m \times m]$ and associated matrices

1: **Step:** DPCCA$(X_t)$ &emsp;&emsp;&emsp;&emsp;&emsp;&emsp;&emsp;&emsp;&emsp;&emsp;&emsp; ▷ Compute pairwise DPCCA
2: &emsp; **for** pair of vectors $\{x^{(a)}\}, \{x^{(b)}\} \in X_t$ **do**
3: &emsp;&emsp; **for** $s$ in $s_{range}$ **do**
4: &emsp;&emsp;&emsp; Compute the coefficient $\rho_{DPCCA}(a, b; s)$ &emsp;&emsp;&emsp;&emsp;&emsp; ▷ Equation(1)
5: &emsp;&emsp; $FC[a, b] \leftarrow \max |\rho_{DPCCA}|$ in $s_{range}$
6: &emsp;&emsp; $P[a, b] \leftarrow$ statistical significance of $FC[a, b]$ given the null empirical distribution
7: &emsp; **return** $FC$ and $P$ &emsp;&emsp;&emsp;&emsp;&emsp;&emsp; ▷ Matrix of connection weights and p-values
8: **Step:** CCA$(X_t)$ &emsp;&emsp;&emsp;&emsp;&emsp;&emsp;&emsp;&emsp;&emsp;&emsp; ▷ Compute CCA connectivity
9: &emsp; **for** $x^{(a)} \in X_t$ **do**
10: &emsp;&emsp; **for** $x^{(b)} \in X_t, b \neq a$ **do**
11: &emsp;&emsp;&emsp; $r_{CCA}[a, b] \leftarrow (1 - \text{CCA between } \{x^{(a)}\}, \{x^{(c)}\}, c \neq a, b)$ &emsp;&emsp; ▷ Effect of excluding node $b$
12: &emsp;&emsp; $index_{con} \leftarrow \text{k-means}(r_{CCA}[a])$ &emsp;&emsp;&emsp; ▷ Split connections into binary groups
13: &emsp;&emsp; $CCA[a, index_{con}] \leftarrow 1$
14: &emsp; **return** $CCA$ &emsp;&emsp;&emsp;&emsp;&emsp;&emsp; ▷ CCA is a binary connectivity matrix
15: **Step:** DPCCA+CCA(P,CCA) &emsp;&emsp;&emsp;&emsp;&emsp; ▷ Augment DPCCA with CCA results
16: &emsp; **for** pair of nodes $\{a, b\}$ **do**
17: &emsp;&emsp; $FC^*[a, b] \leftarrow 1$, if $P[a, b] < 0.05$ &emsp;&emsp;&emsp;&emsp; ▷ DPCCA significant connections
18: &emsp;&emsp; $FC^*[a, b] \leftarrow \max(FC^*[a, b], CCA[a, b])$ &emsp;&emsp;&emsp; ▷ Fill missing connections
19: &emsp; **return** $FC^*$, $FC$ and $P$ &emsp;&emsp;&emsp;&emsp; ▷ $FC^*$ is a binary matrix

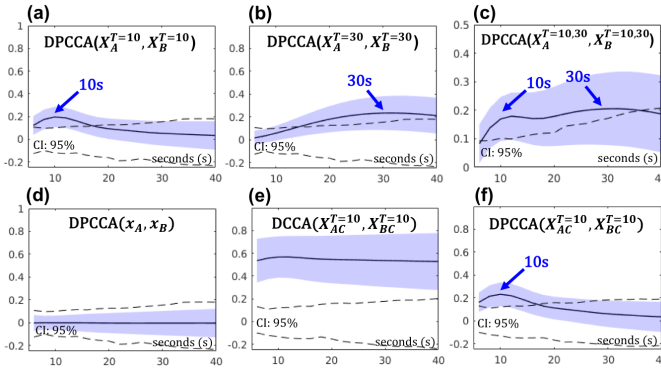

Figure 3: DPCCA temporal profiles among the synthetic signals (details in Section 3.2.1). (a)-(c): DPCCA with peak at T=10s and T=30s, and mixed. (d) DPCCA of the original fMRI signals used to generate the synthetics signals. (e) Temporal profile obtained with DCCA without partial correlation. (f) DPCCA peak at T=10s after controlling for $X_C$. Dashed lines are the 95% confidence interval of DPCCA for the empirical null distribution.

## 4.2 Simulated networks: Improved connectivity accuracy

The goal of this experiment is to validate the proposed methods in an extensive dataset designed to test functional connectivity methods. In this dataset, ground truth networks are known with the architectures aimed to reflect real brain networks. We use the full NetSim dataset comprised of 28 different brain circuits and 50 subjects. For each sample of time series, we compute the partial correlation (parCorr) and the regularized inverse covariance (ICOV), reported as the best performers in [20], as well as the proposed DPCCA and DPCCA+CCA methods. For each measure, we construct empirical null distributions, as described in Section 3.2.2, and generate the binary connectivity matrix using threshold $\alpha = 0.05$. To evaluate their connectivity accuracy, given the ground truth networks, we compute the true positive and negative rates (TPR and TNR, respectively) and the balanced accuracy BAcc=$\frac{(TPR+TNR)}{2}$.

Using NetSim fMRI data as the testing benchmark, we observed that the proposed DPCCA+CCA method provided more accurate functional connectivity results than the best methods reported in the original paper [20]. Results are summarized in Table 1. Here we use the balanced accuracy (BAcc)

as the evaluation metric, since it is a straightforward way to quantify both true positive and negative connections.

Table 1: Comparison of functional connectivity methods using NetSim dataset. Mean and standard deviation of balanced accuracy (BAcc), true positive rate (TPR) and true negative rate (TNR) are reported. ParCorr: partial correlation, ICOV: regularized inverse covariance, DPCCA: detrended cross correlation analysis, DPCCA+CCA: DPCCA augmented with CCA. DPCCA+CCA balanced accuracy is significantly higher than the best competing method ICOV (Wilcoxon signed paired test, Z=3.35 and p=8.1e-04).

| Metrics | Functional connectivity measures | | | | | | | | | | | |
| | ParCorr | | | ICOV | | | DPCCA | | | DPCCA+CCA | | |
| | BAcc | TPR | TNR | BAcc | TPR | TNR | BAcc | TPR | TNR | BAcc | TPR | TNR |
|---|---|---|---|---|---|---|---|---|---|---|---|---|
| Mean | 0.834 | 0.866 | 0.804 | 0.841 | 0.866 | 0.817 | 0.846 | 0.835 | 0.855 | 0.859 | 0.893 | 0.824 |
| Std | 0.096 | 0.129 | 0.188 | 0.095 | 0.131 | 0.181 | 0.095 | 0.150 | 0.177 | 0.091 | 0.081 | 0.169 |

## 4.3   Real-world dataset: Learning connectivity temporal profiles

We use unsupervised methods to (i) learn representative temporal profiles of connectivity from $DPCCA_{Full}$, and (ii) perform dimensionality reduction. The use of temporal profiles may capture additional information (such as short- and long-memory connectivity). However, it increases the feature set dimensionality, imposing additional challenges on classifier training, particularly with small dataset. The first natural choice for this task is principal component analysis (PCA), which can represent original features by their linear combination. Additionally, we use two popular non-linear dimensionality reduction methods Isomap [22] and autoencoders [9]. With Isomap, we attempt to learn the intrinsic geometry (manifold) of the temporal profile data. With autoencoders, we seek to represent the data using restricted Boltzmann machines stacked into layers.

In Figure 4, we show some representative correlation profiles obtained by computing DPPCA among frontoparietal regions (circuit presented in Fig. 1), and the first three principal components. Interestingly, PCA seemed to learn some of the characteristic temporal profiles. For instance, as expected, the first components captured the main trend, while the second components captured some of the short (task-related) and long (resting-state) memory connectivity trends (Figs.4a-b).

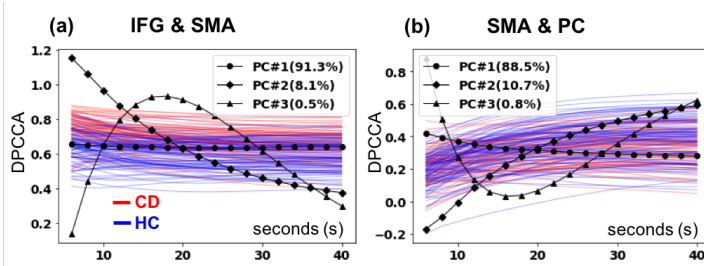

Figure 4: Illustration of some DPCCA profiles and their principal components. IFG: inferior frontal gyrus, SMA: supplementary motor area, PC: premotor cortex. Explained variances of the components are also reported.

## 4.4   Real-world dataset: Cocaine dependence prediction

The classification task consists of predicting the class membership, cocaine dependence (CD) and healthy control (HC), given each individual's fMRI data. After initial preprocessing (Section 2.2), we extract average time series within the frontoparietal circuit of 6 regions [3] (Figure 1), and compute the different cross-correlation measures. These coefficients are used as features to train and test (leave-one-out cross-validation) a set of popular classifiers available in scikit-learn toolbox [16] (version 0.18.1), including k-nearest neighbors (kNN), support vector machine (SVM), multilayer perceptron (MLP), Gaussian processes (GP), naive Bayes (NB) and the ensemble method Adaboost (Ada). For the DPCCA coefficients, we test both peak values $DPCCA_{max}$ as well as the rich temporal profiles $DPCCA_{Full}$. Finally, we also include the brain activation maps (Section 2.2.1) as feature set, thus allowing comparison with popular fMRI classification softwares such as PRONTO (http://www.mlnl.cs.ucl.ac.uk/pronto/). Features are summarized in Table 2.

Table 2: Features used in the cocaine dependence classification task.

| Type | Name | Size | Description |
|---|---|---|---|
| Activation | P(stop) | 1042 | Brain regions responding to anticipation of stop signals |
| | UPE | 1042 | Brain regions responding to unsigned prediction error of P(stop) |
| Connectivity | Corr | 15 | Pearson cross-correlation among the six frontoparietal regions |
| | ParCorr | 15 | Partial cross-correlation among the six frontoparietal regions |
| | ICOV | 15 | Regularized inverse covariance among the six frontoparietal regions |
| | $DPCCA_{max}$ | 15 | Maximum DPCCA within the range 6-40 seconds |
| | $DPCCA_{Full}$ | 270 | Temporal profile of DPCCA within the range 6-40 seconds |
| | $DPCCA_{Iso}$ | 135-180 | Isomap with 9-12 components and 30 neighbors |
| | $DPCCA_{AutoE}$ | 30-45 | Autoencoders with 2-3 hidden layers, 5-20 neurons, batch=100, epoch=1000 |
| | $DPCCA_{PCA}$ | 135-180 | PCA with 9-12 components |

Classification results are summarized in Table 3 and Figure 5. We used the area under curve (AUC) as an evaluation metric in order to consider both sensitivity and specificity of the classifiers, as well as balanced accuracy (BAcc). Here we tested all features described in Table 2, including the DPCCA full profiles after dimensionality reduction (Isomap, autoencoders and PCA). Activation maps produced poor classification results (P(stop): 0.525±0.048 and UPE: 0.509±0.032), comparable to the results obtained with PRONTO software using the same features (accuracy 0.556).

| Features | Mean AUC (± std) | Mean BAcc (± std) | Top classifier (AUC / BAcc) | Accuracy (AUC / BAcc) |
|---|---|---|---|---|
| Corr | 0.757 (± 0.041) | 0.674 (± 0.037) | GP / NB | 0.794 / 0.710 |
| ParCorr | 0.901 (± 0.034) | 0.848 (± 0.025) | GP / Ada | 0.948 / 0.875 |
| ICOV | 0.900 (± 0.030) | 0.838 (± 0.023) | GP / SVM | 0.948 / 0.858 |
| $DPCCA_{max}$ | 0.906 (± 0.019) | 0.831 (± 0.022) | GP / Ada | 0.929 / 0.857 |
| $DPCCA_{Full}$ | 0.899 (± 0.028) | 0.820 (± 0.052) | GP / GP | 0.957 / 0.874 |
| $DPCCA_{Iso}$ | 0.902 (± 0.030) | 0.827 (± 0.068) | GP / MLP | 0.954 / 0.894 |
| $DPCCA_{AutoE}$ | 0.815 (± 0.149) | 0.813 (± 0.106) | SVM / kNN5 | 0.939 / 0.863 |
| $DPCCA_{PCA}$ | 0.928 (± 0.035) | 0.844 (± 0.064) | Ada / NB | 0.963 / 0.911 |

Table 3: Comparison of classification results for different features. The DPCCA features combined with PCA produced the top classifiers according to both criteria (0.963/0.911). However, $DPCCA_{PCA}$ is not statistically better than ParCorr or ICOV (Wilcoxon signed paired test, p>0.05). See Figure 5 for accuracy across different classification methods.

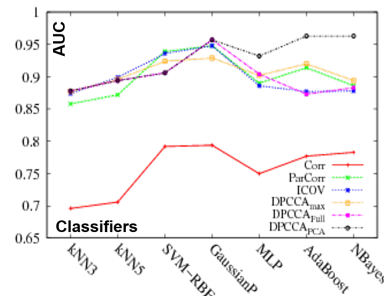

Figure 5: Comparison of classification results for different features and methods (described in Section 4.4).

## 5 Conclusions

In summary, as a multi-time scale approach to characterize brain connectivity, the proposed method (DPCCA+CCA) (i) identified connectivity peak-times (Fig. 3), (ii) produced higher connectivity accuracy than the best competing method ICOV (Table 1), and (iii) distinguished short/long memory connections between brain regions involved in cognitive control (IFC&SMA and SMA&PC) (Fig. 4). Second, using the connectivity weights as features, DPCCA measures combined with PCA produced the highest individual accuracies (Table 3). However, it was not statistically different from the second best feature (ParCorr) across different classifiers. Further separate test set would be necessary to identify the best classifiers. We performed extensive experiments with a large simulated fMRI dataset to validate DPCCA as a promising functional connectivity analytic. On the other hand, our conclusions on clinical prediction (classification task) are still limited to one case. Finally, further optimization of Isomap and autoencoders methods could improve the learning of connectivity temporal profiles produced by DPCCA.

**Acknowledgments**

Supported by FAPESP (2016/21591-5), CNPq (408919/2016-7), NSF (BCS1309260) and NIH (AA021449, DA023248).

## Footnotes

[2]Peak MNI coordinates for IFG:[39,53,-1], MFG:[42,23,38],bilateral insula:[-33,17,8] and [30,20,2], L.PC:[-36,-13,56], and SMA:[-9,-1,50] in mm.

[3]Although these regions are obtained from the whole-group, no class information is used to avoid inflated classification rates.

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
