[Reviews · NeurIPS 2017]

Reviewer 1



In this work, the authors describe the use of detrended partial cross correlation (DPCCA) as a quantity to capture short and long memory connections among brain recordings, for connectivity analysis. DPPCA is complemented with CCA to study the efficacy of detecting connectivity on simulated data (generated with NatSim), and compared to partial correlation and regularized inverse covriance (ICOV). On real fMRI data, DPCCA is first used together with PCA to show representative correlation profiles and perform dimensionality reduction (with Isomap (Iso) and autoencorder (AutoE)). Second, various combinations of DPCCA values and dimensionality reduction methods are used as feature for predicting cocaine dependent class from control. The paper is sufficiently well written and most parts is described in enough detail to reproduce the technical steps of the proposed methodology. I appreciate the use of DPCCA which is definitely new to the neuroimaging data analysis domain. The results provide significant evidence to support the claims. Even thoug the evidence supporting the claims is significant, the effect size of the gain of the proposed method is pretty slim. For example, in Table 3, the authors tested several combinations of methods and reported the "best" classifier in each case. The final result reported in the conclusion comments only the "best of the best" of them, with respect to ICOV. Given the small effect size (AUC=0.963 vs. AUC=0.948), I suspect that the positive difference may be partially due to overfitting. When proposing multiple methods and multiple classifiers, their AUC (or whatever other accuracy measure) is used to rank them and decide the best one. Nevertheless, the best AUC is slightly optimistically biased. In such setting, an unbiased estimate of the AUC of the best method/classifier can be obtained only on a further separate test set - which was not indicated by the authors. Minor issues: - In Section 5, I'd like to see more discussion of the results in the paper (Fig.3, Tab.1, ecc.) and not only a small piece of Tab.3 - The descriptions in Section 4.3 are very dense and should be expanded more, to make them more accessible. Alternatively, some of them could be dropped, since they are not mentioned later on. - Complementing DPCCA with CCA seems an ad-hoc step to show higher results in Table 1. The authors should work more on the motivations of such step. - typo l.43: corrrelation

Reviewer 2



The study presents a method of using DPCCA to discover brain connectivity patterns and differences, which can be used for neuropsychiatric condition prediction (cocaine addict or control). It is an interesting and important approach and the authors basically convince of its usefulness both in artificial and real fMRI data. However, I have a few questions/concerns: - What is the rationale behind choosing the ROIs and would the results hold if other ROIs were chosen? (or for whole brain analyses) - What is the number of PCs in fig.4? It's a bit strange that PC#3 explain only 0.1-0.2% of variance and the first two less than 1/4. - What exactly do results in fig.4 suggest about the interactions between IFG & SMA and SMA & PC? - Finally, are there any neural inferences that could be obtained using DPCCA that are not amenable using other approaches?

Reviewer 3



The paper proposes a partialized version of DCCA, and applies it to synthetic and real fMRI data. While the contribution is incremental (this is a straight analogy to Whittaker's 1990 estimator of partial correlation), the results are at least on par with existing methods, and offer very interesting additional insights into connectivity across time scales. It also brings methods from statistical physics into neuroimaging, where they have been used (for DFA see e.g. e.g. Shen et al Clin Neurophysiol 2003 (EEG), Mutch et al PLoS one 2012 (fMRI)), but not often for connectivity analysis. There are a few technical issues to address. # Section 2.2.1 ROIs are derived from task fMRI on the whole dataset. This is problematic for the cocaine dependence prediction task (section 4.4) because it happens outside the cross-validation fold. The GLM should be re-trained within each fold using only training data, to compute ROIS which would then be applied to constructuct both training and testing graphs. Doing it outside the fold gives an unfair advantage and artificially boosts performance. Given the relatively large (by neuroimaging standards) sample size, this should not be too dramatic but it needs to be acknowledged in the text. This is of course causes no problem in comparisons between connectivity methods since they all use the same ROIs. # Section 3.1 Equation 1: in the denominator, should Ca,bbeCa,a, and likewise for Cb,b ? Is there a guarantee that \rho in equation 2 is always invertible ? # Section 3.2.2, 3.2.3 Why use the max over all time scales rather than some other statistic of the absolute value ? Figure 3 illustrates why the authors chose this, but this choice of an extreme statistic probably comes with a bias towards false positives. In this respect it is not clear why the authors say on lines 177-178 that DPCCA has low TPR? Also, step 17 in algorithm 1 does not correct for multiple comparisons, using an uncorrected p-value while where are many hypothesis tests, which also contributes to false positives. This is not necessarily an issue but, together with the max operator of line 5, it is unclear how to assign proper p-values to each edge after algorithm 1, if that were desireable. The null model chosen works for multi-subject studies only. It yields a very different (much lower) degree distribution. Is this a concern? # Section 4.1 DCCA max is actually also located correctly at 10 s (fig 3e) # Section 4.2 Please move hypothesis test results about DPCCA+CCA > ICOV from conclusion to this section. Given table 1 and large std it is not obvious at all that DPCCA+CCA is better, so having the test result here will be more convincing. # Section 4.4. As mentioned in my remarks on 2.2.1, ROIs are computed out-of-fold, which gives a small unfair edge to all methods, but does not affect comparison between methods computing connectivity on the same ROIs. Using AUC is not recommended for comparing different classifiers (Hand, Machine Learning 77, 2009, but also Hanczar et al., Bioinformatics 26, 2010). Please show balanced accuracy as well. # Section 5 In table 3 it looks as if all methods (except Corr) do about the same, but the authors state in lines 241-242 that DPCCA+PCA works best. Given large std I don't think that the difference with, say ICOV or ParCorr is significant. Please provide a hypothesis test here.